# VGF and Its Derived Peptides in Amyotrophic Lateral Sclerosis

**DOI:** 10.3390/brainsci15040329

**Published:** 2025-03-22

**Authors:** Antonio Luigi Manai, Paola Caria, Barbara Noli, Cristina Contini, Barbara Manconi, Federica Etzi, Cristina Cocco

**Affiliations:** 1NEF-Laboratory, Department of Biomedical Sciences, University of Cagliari, 09042 Cagliari, Italy; 28mansis10@gmail.com (A.L.M.); barbara.noli@unica.it (B.N.); cristina.cocco@unica.it (C.C.); 2Unit of Biology and Genetics, Department of Biomedical Sciences, University of Cagliari, 09042 Cagliari, Italy; federicaetzi@gmail.com; 3Department of Medical Sciences and Public Health, University of Cagliari, 09042 Cagliari, Italy; c.contini@unica.it; 4Department of Life and Environmental Sciences, University of Cagliari, 09042 Cagliari, Italy; bmanconi@unica.it; 5ALS Interdivisional Center, 09042 Cagliari, Italy

**Keywords:** ALS, neuropeptide, proteins, biomarkers, proVGF

## Abstract

Amyotrophic lateral sclerosis (ALS) is a neurodegenerative disease characterized by a progressive degeneration in the neurons of the frontal cortex, spinal cord, and brainstem, altering the correct release of neurotransmitters. The disease affects every muscle in the body and could cause death three to five years after symptoms first occur. There is currently no efficient treatment to stop the disease’s progression. The lack of identification of potential therapeutic strategies is a consequence of the delayed diagnosis due to the absence of accurate ALS early biomarkers. Indeed, neurotransmitters altered in ALS are not measurable in body fluids at quantities that allow for testing, making their use as diagnostic tools a challenge. Contrarily, neuroproteins and neuropeptides are chemical messengers produced and released by neurons, and most of them have the potential to enter bodily fluids. To find out new possible ALS biomarkers, the research of neuropeptides and proteins is intensified using mass spectrometry and biochemical-based assays. Neuropeptides derived from the proVGF precursor protein act as signaling molecules within neurons. ProVGF and its derived peptides are expressed in the nervous and endocrine systems but are also widely distributed in body fluids such as blood, urine, and cerebrospinal fluid, making them viable options as disease biomarkers. To highlight the proVGF and its derived peptides’ major roles as ALS diagnostic biomarkers, this review provides an overview of the VGF peptide alterations in spinal cord and body fluids and outlines the limitations of the reported investigations.

## 1. Introduction

### 1.1. ALS

Amyotrophic lateral sclerosis (ALS), also known as motor neuron disease (MND) or Lou Gehrig’s disease in the United States, is a neurodegenerative disorder resulting in degeneration of both the upper and lower motor neurons that control voluntary muscle contraction [1]. In its initial stage, ALS is characterized by gradual muscle stiffness, twitches, weakness, and wasting [2]. However, motor neuron degeneration increases until eating, speaking, and breathing become difficult [3]. Depending on the developing symptoms, ALS is classified as limb-onset (begins with weakness in the arms or legs) or bulbar-onset (begins with difficulty in speaking or swallowing) [2,4,5]. Despite both genetic and environmental factors being believed to be involved, about 90–95% of ALS cases have unknown causes, named sporadic, while the remaining 5–10% are genetic (hereditary) and known as familial [6]. More than 40 genes have been identified, and among them, four—C9orf72, SOD1, TARDBP, and FUS—account for the disease in up to 70% of people with familial ALS [7].

Compared to 1939, when Lou Gehrig was for the first time diagnosed with ALS, the current diagnosis is not only based on the patient’s symptoms, observed by specialized neurologists [8], but the patients also have access to innovative tests to rule out other conditions or help in diagnosis, including electromyogram (EMG) and magnetic resonance imaging (MRI) [9]. Furthermore, new drugs have been developed and are under confirmatory study, i.e., riluzole and edaravone [10,11], whose mechanism in ALS is uncertain, and tofersen, approved by the FDA in early 2023 for use by people with SOD1-related genetic ALS [12].

However, despite more than 85 years of scientific innovation and novel discoveries of highly sophisticated technologies, there are still no precise and accurate diagnostic approaches in the early phase or drugs that can stop ALS [4]. The fact that ALS is a complex disease has not helped in research progression [4]. Indeed, the complexity is due to the presence of variable phenotypes (classic spinal onset, bulbar, flailed-arm, pseudomyopathic) of the disease [13], the rapidly growing list of environmental factors associated with ALS [14], and finally, ALS mutations, which are associated not only with the genes related to neuron-damaging proteins but also RNA processing, autophagy, vesicular transport, and energy metabolism [15]. The lack of identification of potential therapeutic strategies is also a consequence of the delayed diagnosis due to the absence of accurate ALS early biomarkers [16,17]. Indeed, ALS is linked to an imbalance in neurotransmitters, such as γ-aminobutyric acid [18], acetylcholinesterase [19], and glutamate [20]. However, these neurotransmitters are regrettably not detectable in body fluids in sufficient amounts to enable precise and repeatable testing, which means they are not suitable as diagnostic biomarkers. This happens because neurotransmitters in general are rapidly cleared by specialized transporters from the synaptic cleft; as a result, they do not diffuse far from the site of release and are essentially undetectable in body fluids such as saliva, blood, urine, and cerebrospinal fluid (CSF), which are typically examined in humans for the presence of disease biomarkers. Subsequently, attention has focused on other substances that could be produced by neurons, fluctuate in concert with neurotransmitters, and possibly be found in body fluids.

### 1.2. Neuropeptides

Neuropeptides are chemical messengers that are synthesized and released by neurons and co-released with other neuropeptides or neurotransmitters in a single neuron, yielding a multitude of effects [21]. Neuropeptides are synthesized from inactive precursor proteins [22]. Dense core vesicles are transported throughout the neuron and can release peptides at the synaptic cleft and cell body and along the axons [23]. In contrast to neurotransmitters, which are rapidly removed from the synaptic cleft by specialized transporters, there is no reuptake machinery for peptides [21]. Hence, neuropeptides can diffuse from their point of release and act across a comparatively long distance since they are only gradually eliminated from the extracellular space [24]. The combination of volume transmission and lack of reuptake contributes to the relatively long-lasting effects of neuropeptides [21,24]. As a result, neuropeptides are long-lived, and inactivation occurs by extracellular proteases, which in some cases can even generate new bioactive peptides by cleaving existing neuropeptides. There are at least 100 neuropeptides in the brain, and some of them are released in the blood with a hormone function [21,25]. There are several ways to assess the number of peptides or proteins in biological fluids, but the most popular ones are immunoassays (EIA/ELISA/SIMOA) [26,27,28]. All these immunoassay techniques are based on antigen-antibody reactions and offer fast throughput, automated reruns, relatively inexpensive tests of high sensitivity and specificity, and results that can be reported in a routine format. However, like every analytical method, immunoassays also suffer from some limitations. Interferences and pitfalls are related to the use of antibodies, i.e., unspecific cross-reactivity that could make it difficult to discriminate between similar proteins or peptides. Advancing technologies, particularly liquid chromatography (LC) coupled to high-resolution tandem mass spectrometry (HR-MS/MS), have radically improved the precision of identifying and measuring proteins and peptides in biological samples [29]. Mass spectrometry-based proteomic studies can be performed through two approaches, i.e., bottom-up and top-down. The bottom-up approach involves enzymatic digestion (often trypsin) of the sample proteins prior to the analysis [30] and allows analyzing complex samples, providing extensive characterization of the proteome profiles under study. However, protein sequence coverage could be insufficient, making it impossible to reconstruct information about genetic variation or alteration within the context of the entire protein. On the other hand, the top-down analysis provides higher sequence coverage, valuable deciphering of post-translational modifications, and improved quantification compared to the bottom-up method [31]. In fact, this latter method is crucial for clarifying the precise proteolytic processing mechanisms that transform precursor proteins into small peptides, making it especially pertinent to neuroproteomics. [32]. However, compared to the immunoassay approach, a limitation of mass spectrometry-based proteomics is the high skill required to use sophisticated instruments and software; therefore, previous results with alternative methods could be valuable to guide the proteome investigation. To search for novel biomarkers, the results by mass spectrometry methods should be validated with other approaches, such as immunoassays, which can then be applied to large-scale population screening.

### 1.3. VGF Precursor Protein and Its Derived Peptides

The neurosecretory protein VGF was first identified in the VGF7a/8a gene (human and mouse, respectively) from PC12 cells [33]. VGF mRNA is potently upregulated by many factors, including brain-derived neurotrophic factor (BDNF) [34], neurotrophin-3 (NT-3) [35], and nerve growth factor (NGF) [36]. Full-length human VGF1-615 (or rat/mouse VGF1-617) is mainly synthesized in neurons of the central (including cortex and spinal cord) and peripheral nervous systems [37,38,39,40] as well as neuroendocrine cells [41,42,43,44]. The proVGF undergoes proteolytic processing in the regulated secretory pathway to produce several VGF-derived peptides. VGF-derived peptides are named by their first N-terminus amino acids and the total length; hence, among the small VGF-derived peptides are the TLQP-21 [45,46,47], AQEE-30 [48], and TLQP-62 [49], while among the large VGF-derived peptides is the NAPP-129 (VGF485–615), which includes TLQP-62 and consists of 131 amino acids in humans (129 amino acids in murine species) [49]. VGF-derived peptides are conserved among humans and other mammals, including rats, mice, and chimpanzees, highlighting their crucial role in normal physiology throughout evolution [50]. In the nervous system, VGF-derived peptides act as signaling molecules [51] and are involved in a range of processes for regulating energy balance [52,53] and neurogenesis [54], as well as learning and memory [55]. VGF peptides also have a protective role. In fact, the AQEE-30 prevents cell damage in Huntington’s disease animal models [56] and has been linked to exercise-mediated BDNF increase [57]. Furthermore, in a transgenic mouse model of Alzheimer’s disease (AD), the binding of the TLQP-21 reduced amyloid-β (Aβ) plaques and neurite dystrophy [58].

To date, several analytical platforms, among which mass spectrometry is the most represented, have allowed discovering proVGF-derived fragments not only in neurons [59] or cell lines [60] but also in different biological matrices, such as CSF [61,62,63], urine [64], and blood [65]. The hypothesis that VGF and its derived neuropeptides may play a role in the context of different neurodegenerative diseases is supported by the evidence of altered levels of specific proVGF-derived fragments in CSF from dementia with Lewy bodies (DLB) [63], Parkinson’s disease (PD) [65], AD [66], multiple sclerosis [67], and ALS [68,69,70]. In some cases, CSF alterations of VGF-derived peptides revealed by mass spectrometry have been ensured by using immuno-enzymatic assays with specific VGF peptide antibodies, proving the high potential of specific VGF peptides as good biomarkers [63]. To highlight the proVGF and its derived peptides’ major roles as ALS diagnostic biomarkers, this review provides an overview of the VGF alterations in spinal cord and body fluids related to ALS and outlines the limitations of the reported investigations.

### 1.4. Methods

In order to conduct this review, the term VGF was searched on PubMed either by itself or in conjunction with ALS and neurodegeneration.

## 2. VGF Expression and Changes in ALS Nervous System

ProVGF and its related peptides are expressed in the spinal cord of both human and animal models (mice) and have been shown to exhibit changes related to ALS. Only one study reveals a VGF decrease in the spinal cord of ALS patients. Reduction in the dorsal and anterior horns of the spinal cord (9 patients vs. 9 controls) was observed through immunohistochemistry using a non-identified VGF antibody and in situ hybridization [71]. Interestingly, individuals with ALS who survived for an extended period (*n* = 3) maintained the same amount of VGF protein expression [71]. In terms of the animal research, the ALS animal model used in each of the studies detailed here was SOD1-G93A transgenic mice. It must be noted that this model, like the majority of the ALS animal models, lacks important neurodegenerative characteristics, which poses difficulties and restrictions for its application [72]. VGF immunoreactivity reduction in the ALS animal model was observed using one VGF commercial antibody, referred to as the antibody against the VGF full length; however, the molecular weight (MW) forms it recognized were not investigated [70]. Using this antibody, the VGF immunoreactivity was found in the same spinal cord perikarya containing the small integral membrane protein 32 (SMI-32) but not Neu-N or the glial fibrillary acidic protein. VGF was observed already in asymptomatic animals, but it continued to decline as animals weakened [70]. Interestingly, the loss of VGF immunoreactivity overlaps quantitatively with the loss of SMI-32 [70]. Two additional VGF antibodies, distinct from the one previously mentioned, were employed on spinal cord sections of the same mice in two further published studies [40,49]. One antibody was directed against the proVGF C-terminus portion (9 amino acids), and it will be further referred to as the VGF C-terminus antibody [49], while the other was specifically produced against the TLQP-9 peptide [40]. This latter antibody will be further referred to as the TLQP antibody. In the spinal cord of control mice, the VGF C-terminus antibody labeled the ventral and dorsal horns of the spinal cord, where almost all the VGF-positive cell bodies were identified as motor neurons using the vesicular acetylcholine transporter (VAChT) [49]. To better identify which MW forms were labeled by the VGF C-terminus antibody, spinal cord extracts of control mice were used with either gel chromatography coupled with ELISA or HPLC-ESI-MS. Using these approaches, it was found that VGF C-terminus antibodies labeled proVGF (of 66 kDa), NAPP-129 (14–15 kDa), and other peptides of 1.54 and 2.5 kDa, namely AQEE-13 and NAPP-19, and a novel peptide of 2.5 kDa called ELQE-20 [49]. VGF C-terminus immunoreactivity was reduced already at the presymptomatic stage of motor neurodegeneration, and it was equally slight in the late-symptomatic stage while the VAChT labeling remained visible [49]. Similar cell localization and alteration profiles (i.e., decrease at the pre-symptomatic stage in motor neurons) were obtained using the TLQP antibody [40]. However, chromatography analysis coupled with ELISA in extracts of the control spinal cord revealed that the antibody recognized not only high MW forms compatible with the proVGF and NAPP-129 (the same labeled by the C-terminus antibody) but also other smaller peptides of ~7, 5, 3, and 2 kDa compatible with TLQP-62, -42, -30, and -21, respectively [40]. The specific VGF fragments changed in the spinal cord are summarized in Figure 1.

## 3. VGF Changes in CSF, Serum, and Plasma of ALS Patients

To date, VGF changes (reduction) in ALS body fluids have been reported in CSF, serum, and plasma.

### 3.1. VGF Changes in CSF

In 2006, the first proteomic analysis of CSF samples from ALS patients demonstrated that a VGF fragment of 4.8 kDa (ARQNALLFAEEED or ARQN-13 at proVGF398–410 sequence) was able to discriminate (through its decrease) with high sensitivity and specificity between patients with ALS (*n* = 49) and patients with other neurologic disorders (*n* = 5; multifocal motor peripheral neuropathy; *n* = 2; sensorimotor peripheral neuropathy) or normal subjects (*n* = 46) analyzed using the surface-enhanced laser desorption/ionization time-of-flight mass spectrometry (SELDI-MS) proteomics technique [73]. However, fragments containing the ARQNALLFAEEED amino-acid sequence were also changed in the CSF samples of patients affected by acute encephalopathy [69] and AD [61,66,74,75,76]. In 2008, using an ELISA sandwich with a capture antibody against the C-terminal epitope of proVGF (588–617) and a detection antibody against the proVGF 78–340 sequence, VGF levels were found to be decreased in the CSF of patients with ALS (17 vs. 21 controls) [70]. Using this approach, the VGF decrease correctly diagnosed ALS patients with 77% sensitivity and 87% specificity based on receiving operating characteristic (ROC) analysis [70]. Studies in 2020 [77] and 2024 [68] revealed further VGF fragments decreased in ALS-CSF. The 2020 study identified the CSF presence of the AQEEAEAEER or AQEE-10 (proVGF586-595) by analyzing samples from ALS and frontotemporal dementia (FTD) patients, both with hexanucleotide repeat expansion in the C9orf72 gene, which is the most common mutation associated with ALS (C9-ALS) and frontotemporal dementia (C9-FTD). Isobaric tags for relative and absolute quantitation (iTRAQ) and liquid chromatography/tandem mass spectrometry (LC-MS/MS) were used [77], revealing a significant increase in AQEE-10 peptide in patients affected by C9-ALS (*n* = 16) compared to C9-FTD patients (*n* = 9). However, the authors of the study did not validate the iTRAQ results using MRM analysis [77]. In 2024, three further VGF fragments [68], PPGRPEAQPPPLSSEHKEPVAGDAVPGPKDGSAPEVRGA, AVPGPKDGSAPEV, and APPEPVPPPRAAPAPT, or PPGR-39, AVPG-13, and (N)APPE-19, at proVGF 24–62, 47–59, and 487–504, respectively, were decreased in CSF samples from ALS patients (*n* = 50) compared to non-ALS controls (*n* = 50; patients without neurodegenerative diseases who underwent lumbar puncture for differential diagnosis of facial palsy or headaches). Probably due to the shortage of CSF amount that could be obtainable in mice, only one study [70] reported a VGF decrease in the CSF of SOD1 G93A transgenic mice. Using the RIA technique with an antibody to the AQEE-30 peptide, VGF was decreased as a function of the progression of muscle weakness characterized by an increasing number of affected muscle segments, assessed by manual muscle testing. Interestingly, CSF-VGF reduction was found to precede the onset of muscle weakness assessed by the rotarod assay. At the end of this section, there are some points to consider. One is that the VGF fragments found using mass spectrometry in humans were revealed exclusively by the bottom-up methodology using digestion with trypsin. As a result, it is still unclear whether the variations in the VGF peptides were due to individual native peptides or whether modifications in the digested fragments were due to a change in the VGF precursor. Second, although CSF is the main reservoir for the impacts of neuronal changes during pathological processes [78,79] and a significant source of potential biomarkers [78], lumbar punctures are frequently challenging to execute in ALS patients because they are associated with several consequences (loss of mobility, overall fragility, headache) [79]. The specific VGF sequences changed in CSF are summarized in Figure 2.

### 3.2. VGF Changes in Serum and Plasma

A VGF decrease was revealed in blood (serum or palsma) from both ALS patients and G93A SOD-1 mice [40,49,70]. In ALS patients, using competitive ELISA with a VGF C-terminus antibody, VGF reduction was observed only in patients with advanced ALS (*n* = 18 vs. 23 patients with early ALS and 45 controls) [49]. Interestingly, competitive ELISA using the same antibody revealed VGF reduction in plasma from patients with initial PD [65], indicating that VGF C-terminal peptides are more valuable as early indicators for PD than ALS. In control plasma samples, using gel chromatography in conjunction with ELISA, the VGF C-terminus antibody was able to label the proVGF and the NAPP-129. Using the same ALS patients but competitive ELISA with the TLQP antibody, VGF reduction was instead seen in the initial ALS phase [40]. In control plasma samples, using gel chromatography in conjunction with ELISA, the TLQP antibody was able to recognize the same forms revealed with the VGF C-terminus antibody plus TLQP-21, TLQP-30, TLQP-42, and TLQP-62 [40]. Animal models (SOD-1 mice) were studied using ELISA or RIA methods [40,49,70]. Using RIA with an AQEE-30 antibody, serum reduction (in parallel with CSF decrease) was found to precede the onset of muscle weakness assessed by rotarod assay relative to age-, gender-, and age-matched wild-type littermates [70]. However, no qualitative or quantitative examination of the specific VGF peptide sequences labeled by the antibody was carried out. TLQP and VGF C-terminus antibodies were also employed using plasma samples from the same animal model, revealing a decrease in VGF levels, which were associated with the asymptomatic [40] and late ALS phases [49], respectively. The identical MW forms observed in humans were detected with both antibodies in control mouse plasma; however, two additional forms that corresponded to TLQP-62 and AQEE-30 were found with the VGF C-terminus antibody [49]. In conclusion, only immunoassays were used to reveal blood VGF changes. The MW forms recognized by the VGF antibodies were only analyzed in control samples and in certain cases. Hence, it is still unknown which VGF peptides among those labeled by the antibodies are truly decreased in ALS blood samples. The specific VGF fragments changed in body fluids (serum and plasma) are summarized in Figure 3.

## 4. VGF Role in ALS Mechanisms

It has to be reported at this point that the majority of the VGF peptide fragments found by mass spectrometry have not been identified for their biological activities; hence, the majority of the VGF peptides revealed until now remain uncharacterized. The etiology of ALS may be linked to pathways resulting in VGF decrease, as the loss of certain VGF peptides (AQEE-30) and/or total full-length VGF are associated with a progressive deterioration of muscles [70]. On the other hand, increasing VGF might have a therapeutic effect, as shown using primary mixed spinal cord neuron cultures, in which exogenous proVGF lowers excitotoxic damage [70]. Accordingly, in another in vitro study [80], the protective effect of SUN N8075, a radical scavenger with neuroprotective activity, may be mediated by upregulation of VGF mRNA and attenuated by VGF decrease (achieved via siRNA). The effects of SUN N8075 on motor dysfunction and survival were also reported in SOD-1 animal models [80]. Another research study provided evidence that one VGF-derived peptide (the TLQP-21) has neuroprotective function [40]. In fact, according to the TLQP decrease observed in the plasma and spinal cord, TLQP levels were also reduced in the NSC-34 cell line treated with sodium arsenite (to induce oxidative stress); however, when TLQP-21 was added to these cells, it increased cell viability. Nevertheless, this result was not confirmed by another study that examined the neuroprotective effects of TLQP-21 and additional VGF-derived peptides (AQEE-30, AQEE-11, LQEQ-19, QEEL-16, LENY-13, and HVLL-7) using NSC-34 with SOD1-G93A mutation and serum deprivation [81]. This study revealed that AQEE-30 and LQEQ-19, but not TLQP-21, protected against toxicity and that LQEQ-19 did this by increasing phosphorylation via the Akt and ERK1/2 proteins [81].

Interestingly, AQEE-30 is involved in synapsis [34] and inflammatory [67] mechanisms; TLQP-21 modulates microglial function through C3aR1 signaling pathways and reduces neuropathology in mice [58], while LQEQ-19 has not been well characterized for its biological activity.

In conclusion, it seems that the VGF precursor and the peptides it produces could have a role in the pathophysiology of ALS, and the VGF peptide TLQP-21 could protect the cells against oxidative stress mechanisms; however, further investigations are needed to confirm it.

## 5. Discussion

This review emphasizes the possibility that the peptides produced by the VGF precursor may have a role as early ALS biomarkers. Diagnostic biomarkers should change in such a way that they manifest in the body’s peripheral organs, indicating a disease in the brain. According to multiple studies, VGF alterations have been found in ALS not only within the spinal cord (ventral horn) but also in CSF and blood (plasma or serum). In the spinal cord, in situ hybridization verifies the existence and modifications of VGF mRNA, where immunohistochemistry identifies the specific cells revealing changes in proVGF and its peptides, whereas ELISA, coupled with HPLC-MS or gel chromatography, was used to characterize the antibodies and identify the changes. To identify VGF peptides and quantify the VGF changes occurring in body fluids, mass spectrometry and immunoassays (ELISA or RIA) coupled with gel chromatography were used. The literature on the decreased VGF sequences in ALS is summarized in Table 1. Nevertheless, the investigations have certain drawbacks. For instance, it remains to be ascertained whether VGF changes are unique to ALS, because other neurodegenerative disorders were not extensively studied. Moreover, none of the VGF peptides thought to be changed in the spinal cord or blood have been specifically identified, because the changes revealed with immunoassays (competitive ELISA or RIA) were not ensured by other techniques. Indeed, when VGF alterations are detected using antibodies, the likelihood that a VGF antibody produced against one of the VGF peptides may also detect the others makes it difficult to determine which peptide is changed from the others. Indeed, even though VGF peptides originate from the same precursor protein, their bioactivity, production, or mechanism of action may differ. This is the case for TLQP-21 [82,83] and TLQP-62 [40], which could both be revealed by the same TLQP antibody but have distinct bioactivities when tested in parallel [84]. Another example of how crucial it is to identify VGF peptides correctly is the behavior of the VGF peptides altered in ALS and PD patients’ blood. These studies suggest a major involvement of TLQP peptides in early ALS [40] and of VGF C-terminus peptides in early PD [as well as advanced stages of both ALS and PD (but not early ALS)] [49,65,85,86].

Conversely, it has been determined which specific VGF peptide sequence is changed in ALS-CSF. Despite this accomplishment, two things need to be taken into account. One is that CSF-VGF identification has only been performed by the use of the bottom-up approach; hence, it is still unclear if these fragments are endogenous peptides or the result of digestion. Second, although CSF is a significant source of potential biomarkers [78], lumbar punctures are difficult to perform in ALS patients [79]. For this reason, other peripheral tissues, such as blood (plasma or serum), are favored in the search for a valuable biomarker in ALS.

However, whereas immunoassays can be easily employed to detect VGF peptides in blood samples, mass spectrometry has a number of drawbacks that contribute to the paucity of research in this field. Due to the heterogeneous bioavailability and detectability of VGF peptides, plasma remains a huge complex matrix to be analyzed by these approaches. Indeed, the analysis of VGF peptides using mass spectrometry with plasma samples is hampered by the dynamic range of the samples and the masking effect and interference caused by abundant proteins like albumin, immunoglobulins, haptoglobin, transferrin, and α-1-antitrypsin, which together represent up to 85% of the total protein [87]. Additionally, plasma volume and body mass index may influence the overall concentration of peptides, and complexity is added by the formation of hetero-aggregates and immunocomplexes driven by endogenous humoral responses. Hence, it is important to underline the necessity to employ specific protocols to deplete high-abundant proteins as well as to enrich the low-abundant peptides of interest to maximize the chance of measuring VGF peptides using mass spectrometry with plasma samples [87].

## 6. Conclusions

In conclusion, although VGF peptides are altered in a variety of disorders, the kinds of modified VGF peptides may differ according to the disease condition. Therefore, to highlight specific VGF-derived peptides as early ALS diagnostic biomarkers, it is essential to precisely identify the VGF sequences that change in ALS.

## Figures and Tables

**Figure 1 brainsci-15-00329-f001:**
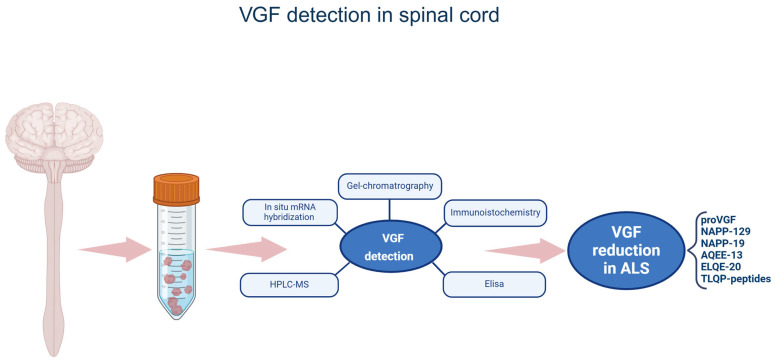
VGF detection in spinal cord. The figure summarizes the VGF sequences identified in human and mouse and reduced in the spinal cords of ALS samples, along with the techniques used. VGF-derived peptides are named by their first N-terminus amino acids and the total lengths. Elisa: enzyme-linked immunosorbent assay. The figure was created in https://BioRender.com.

**Figure 2 brainsci-15-00329-f002:**
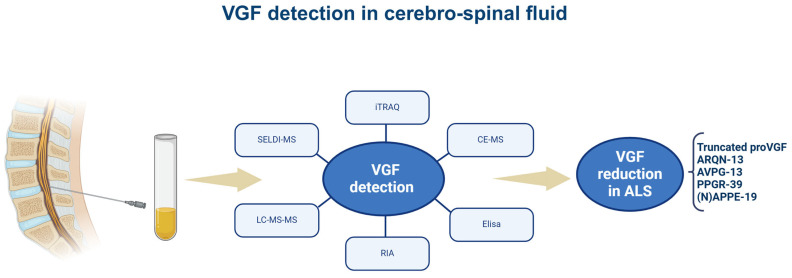
VGF detection in cerebrospinal fluid. The figure summarizes the VGF sequences identified in human and mouse and reduced in the spinal cords of ALS samples as well as the techniques used. VGF-derived peptides are named by their first N-terminus amino acids and the total lengths. ELISA: enzyme-linked immunosorbent assay RIA: radioimmunoassay; SELDI-MS: surface-enhanced laser desorption/ionization–mass spectrometry; CE-MS: capillary electrophoresis–mass spectrometry; LC-MS-MS: liquid chromatography with tandem mass spectrometry, iTRAQ: Isobaric Tag for Relative and Absolute Quantitation. The figure was created in https://BioRender.com.

**Figure 3 brainsci-15-00329-f003:**
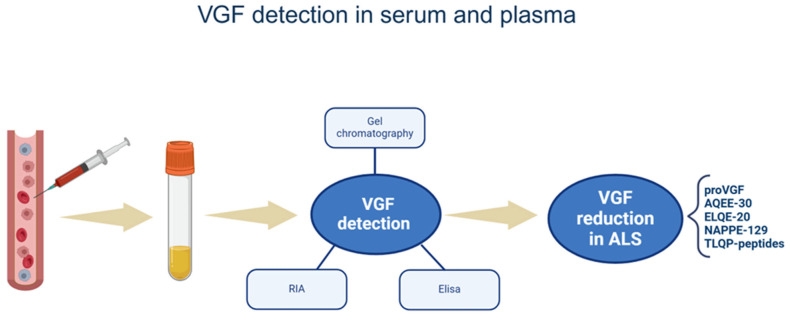
VGF detection in serum and plasma. The figure summarizes the reduction of VGF sequences in ALS animals’/patients’ body fluids, such as serum and plasma, as well as the techniques used. VGF-derived peptides are named by their first N-terminus amino acids and the total lengths. RIA: radioimmunoassay; ELISA: enzyme-linked immunosorbent assay. The figure was created in https://BioRender.com.

**Table 1 brainsci-15-00329-t001:** The literature on the decreased VGF sequences in ALS samples is summarized in the table, along with the methods employed (including antibodies named as they are referred to in the papers). VGF-derived peptides are named by their first N-terminus amino acids and the number of total lengths. SEC: size-exclusion gel chromatography; HPLC-MS: High-performance liquid chromatography-mass spectrometry; ELISA: enzyme-linked immunosorbent assay; SELDI-MS: surface-enhanced laser desorption/ionization mass spectrometry; CE-MS: capillary electrophoresis-mass spectrometry; LC-MS/MS: liquid chromatography with tandem mass spectrometry; RIA: radioimmunoassay; h: human; m: mouse.

Reference	Method	Antibody	VGF Peptide/Protein	VGF Sequence	Samples
[71]	Immunohistochemistry	Non-identified	proVGF	VGF_1–617_	h. spinal cord
[49]	SEC/HPLC-MS/ELISA Immunohistochemistry	VGF C-terminus	proVGF	VGF_1–617_	
			NAPP-129	VGF_486–617_	
			NAPP-19	VGF_486–504_	
			AQEE-13	VGF_586–602_	m. spinal cord
			ELQE-20	VGF_353–372_	
[40]	SEC/ELISA	TLQP	proVGF	VGF_1–617_	
	Immunohistochemistry		NAPP-129	VGF_486–617_	
			TLQP-21, -30, -42, -62	VGF_556–577/556–587/556–598/556–617_	
[73]	SELDI-MS	-	ARQN-13	VGF_398–410_	
[70]	ELISA sandwich	VGF_588–617_ and VGF_78–340_	proVGF	VGF_78–617_	
[68]	CE-MS, LC-MS/MS	-	PPGR-39	VGF_24–62_	h. CSF
			AVPG-13 (N)APPE-19	VGF_47–59_ VGF_487–504_	
[70]	RIA	AQEE-30	AQEE-30	VGF_586–617_	m. CSF
[49]	SEC/ELISA	VGF C-terminus	proVGF	VGF_1–617_	
			NAPP-129	VGF_486–617_	
[40]	SEC/ELISA	TLQP	proVGF	VGF_1–617_	h. blood
			NAPP-129	VGF_486–617_	
			TLQP-21, -30, -42, -62	VGF_556–577/556–587/556–598/556–617_	
[70]	RIA	AQEE-30	AQEE-30	VGF_586–617_	
[49]	SEC/ELISA	VGF C-terminus	proVGF	VGF_1–617_	
			NAPP-129	VGF_486–617_	
			TLQP-62	VGF_556–617_	m. blood
			AQEE-30	VGF_586–617_	
[40]	ELISA	TLQP	proVGF		
			NAPP-129	VGF_486–617_	
			TLQP-21, -30, -42, -62	VGF_556–577/556–587/556–598/556–617_	

## Data Availability

No new data were created by the authors of this scoping review, which summarized the existing evidence from the scientific literature.

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
