# Peer review of "VGF and Its Derived Peptides in Amyotrophic Lateral Sclerosis"

_brainsci, 2025, doi:10.3390/brainsci15040329_

Round 1
Reviewer 1 Report
Comments and Suggestions for Authors
Thank you for giving me the opportunity to review this important manuscript.
The topic is very relevant due to the lack of research into the effectiveness of determining VGF and its derived peptides in the diagnosis and prognosis of patients with ALS.
The title of the study matches the content.
In general, the introduction is written in a very interesting and highly scientific manner, using the latest scientific achievements in this field. However, I have some comments.
1. In lines 49-50: “The ALS diagnosis is mainly based on a patient’s symptoms observed by specialized neurologists”. In my opinion, it is necessary to indicate the decisive role of needle and stimulating electroneuromyography.
2. In lines 50-52 : “Unfortunately, people diagnosed now do not have access to better diagnosis than those available in 1939, when Lou Gehrig was for the first time diagnosed with ALS in June of 1939”. This phrase needs to be corrected. Over the past 30 years, many diagnostic capabilities have been introduced, including high-resolution MRI studies, evoked potentials, and high-precision electroneuromyography with greater capabilities.
3. Add references please to lines: 52, 52, 55, 57, 63, 67
Add purpose for your study.
Please add a separate section for materials and indicate methods for compiling this review, indicating keywords and search engines.
2.VGF expression and changes in the ALS Nervous System. Thanks to the authors for such detailed information and knowledge that they showed us in this section. However, a large flow of information using many terms reduces the clarity and readability of this manuscript. It might be better if you used tables or figures to show your results.
3. VGF changes in body fluids of ALS patients. In my opinion it will be more informative if you changed the title of this section to “VGF changes in CSF and plasma of ALS patients”.
Add references to lines 237.
3.2.VGF changes in blood. Change the title and in the text to “VGF changes in plasma” in accordance with the above term in line 199.
I think it would be better if you add a table or figure to these sections. 3.1. and 3.2.
4.VGF role in ALS mechanisms. In this section, the authors demonstrate recent evidence for the pathogenesis of VGF and its derived peptides in development and progression amyotrophic lateral sclerosis.
The conclusion is too long. I think this section should be better for discussion. It contains all the received data. You can draw conclusions below in a few specific phrases based on the results obtained.
Move table 1 to discussion section.
The references are not subject to journal rules.
Author Response
Thank you for giving me the opportunity to review this important manuscript.
Answer: The referee's remarks have been greatly appreciated. As a result, we have made the necessary adjustments, which are now highlighted in yellow.
The topic is very relevant due to the lack of research into the effectiveness of determining VGF and its derived peptides in the diagnosis and prognosis of patients with ALS.
The title of the study matches the content.
In general, the introduction is written in a very interesting and highly scientific manner, using the latest scientific achievements in this field. However, I have some comments.
In lines 49-50: “The ALS diagnosis is mainly based on a patient’s symptoms observed by specialized neurologists”. In my opinion, it is necessary to indicate the decisive role of needle and stimulating electroneuromyography.
Answer: We have adjusted the sentence appropriately.
In lines 50-52 : “Unfortunately, people diagnosed now do not have access to better diagnosis than those available in 1939, when Lou Gehrig was for the first time diagnosed with ALS in June of 1939”. This phrase needs to be corrected. Over the past 30 years, many diagnostic capabilities have been introduced, including high-resolution MRI studies, evoked potentials, and high-precision electroneuromyography with greater capabilities.
Answer: as above, we have adjusted the sentence appropriately
Add references please to lines: 52, 52, 55, 57, 63, 67
Answer: We have changed the section and adjusted the references appropriately.
Add purpose for your study
Answer: We have added the purpose of the study at the end of Section 1.3.
Please add a separate section for materials and indicate methods for compiling this review, indicating keywords and search engines.
Answer: A sentence has been added at the end of the introduction section (1.4).
VGF expression and changes in the ALS Nervous System. Thanks to the authors for such detailed information and knowledge that they showed us in this section. However, a large flow of information using many terms reduces the clarity and readability of this manuscript. It might be better if you used tables or figures to show your results.
Answer: We appreciate the referee's feedback, and we have added a new figure explaining the results in the nervous system.
VGF changes in body fluids of ALS patients. In my opinion it will be more informative if you changed the title of this section to “VGF changes in CSF and plasma of ALS patients”.
Answer: Since the section includes studies with the use of both serum and plasma, we have modified the title to "VGF changes in CSF, serum, and plasma of ALS patients."
Add references to lines 237.
Answer: We have added the references appropriately.
VGF changes in blood. Change the title and in the text to “VGF changes in plasma” in accordance with the above term in line 199.
Answer: again, since the section includes studies with the use of both serum and plasma, we have modified the title to "VGF changes in serum and plasma "
I think it would be better if you add a table or figure to these sections. 3.1. and 3.2.
Answer: We appreciate the referee's feedback, and we have added two figures explaining the results in the CSF, serum and plasma.
VGF role in ALS mechanisms. In this section, the authors demonstrate recent evidence for the pathogenesis of VGF and its derived peptides in development and progression amyotrophic lateral sclerosis.
The conclusion is too long. I think this section should be better for discussion. It contains all the received data. You can draw conclusions below in a few specific phrases based on the results obtained.
Answer: There are now "discussion" and "conclusion" parts.
Move table 1 to the discussion section.
Answer: Table 1 has been relocated to the new "discussion" section
The references are not subject to journal rules.
Answer: We hope now they are adjusted appropriately.
Reviewer 2 Report
Comments and Suggestions for Authors
This paper is reviewing studies (including those of the authors - 6 refs) on VGF as a biomarker and neuroprotector in ALS. Although VGF as ALS biomarker came into focus in the first decade of the century a novelty is its finding (i.e. its fragments) in the blood thus allowing for more feasible diagnostics. Many molecular markers deriving from VGF are mentioned but their physiological significance is scarcely explained.
A few lines need to be corrected/supplemented in order to make this published.
Line 30: a review is not really a “study”
Lines 52-55: It is an overstatement that that there is a “slow progress” and a “slow research” in the field of ALS neuroscience. In this respect it is true that there are no drugs that can stop ALS but the authors should acknowledge the drugs that modify the disease progress including the ASO technology.
Line 57: Please explain why ALS is “heterogeneous and complex”
Lines 60-64: Glutamate is not mentioned and it is the main driver of excitotoxicity in ALS. In fact it IS found in patient CSF (e.g. PMID: 2375629, PMID: 11790386).
Lines 146-147: If all these diseases have VGF as a marker what is then the specificity of this marker for ALS. Please elaborate.
Line 179: TLQP-9 is not introduced in chapter 1.3. Why? Why is it presented separately?
Lines 194-196: Is there any phenotypic/mechanistic/predictive significance to all these peptides?
Line 228: “non-ALS controls” is not sufficient. Mention which.
Lines 259 and 283: Which SOD1 mutant?
Lines 283-284: Which “VGF-derived peptide”?
Line 286: The authors should acknowledge this is not really an ALS model.
Line 293-295: Can you elaborate? Anything more concrete on mechanism?
Line 297: Not really Conclusions. Rather a Discussion.
Line 49: “on a patient’s symptoms”
Line 90: “routinary” – “routine”
Line 129: should be plural “peptides”
Line 134: “synapses”
Line 204: “sensitivity” instead of “sensibility”
Lines 280 and 282: SUNN8075 or SUN N8075?
Line 298: “emphasizes” should stand better than “raises”.
Comments on the Quality of English Language
Line 49: “on a patient’s symptoms”
Line 90: “routinary” – “routine”
Line 129: should be plural “peptides”
Line 134: “synapses”
Line 204: “sensitivity” instead of “sensibility”
Lines 280 and 282: SUNN8075 or SUN N8075?
Line 298: “emphasizes” should stand better than “raises”.
Author Response
This paper is reviewing studies (including those of the authors - 6 refs) on VGF as a biomarker and neuroprotector in ALS. Although VGF as ALS biomarker came into focus in the first decade of the century a novelty is its finding (i.e. its fragments) in the blood thus allowing for more feasible diagnostics. Many molecular markers deriving from VGF are mentioned, but their physiological significance is scarcely explained.
Answer: The referee's remarks have been greatly appreciated. As a result, we have made the necessary adjustments, which are now highlighted in yellow.
Regarding the specific request, the referee is right; unfortunately, almost all of the VGF peptide fragments found by spectrometry methods have not yet had their biological activity investigated. These points have been added to the manuscript.
A few lines need to be corrected/supplemented in order to make this published.
Line 30: a review is not really a “study”
Answer: We now use the word "review" instead of "study."
Lines 52-55: It is an overstatement that that there is a “slow progress” and a “slow research” in the field of ALS neuroscience. In this respect it is true that there are no drugs that can stop ALS but the authors should acknowledge the drugs that modify the disease progress including the ASO technology.
Answer: We appreciate the referee's feedback and have adjusted the manuscript appropriately.
Line 57: Please explain why ALS is “heterogeneous and complex”
Answer: The complexity is due to the presence of variable phenotypes (classic spinal onset, bulbar, flailed-arm, pseudomyopathic) of the disease, the rapid-growing list of environmental factors associated to ALS, and finally, the mutations, which are associated not only with the genes related to neuron-damaging proteins but also RNA processing, autophagy, vesicular transport, and energy metabolism. We have added this part to the manuscript, including the relevant references.
Lines 60-64: Glutamate is not mentioned, and it is the main driver of excitotoxicity in ALS. In fact it IS found in patient CSF (e.g. PMID: 2375629, PMID: 11790386).
Answer: We appreciate the referee's feedback and have adjusted the manuscript mentioning glutamate.
Lines 146-147: If all these diseases have VGF as a marker, what is then the specificity of this marker for ALS. Please elaborate.
Answer:
As we have better explained in the new version of the manuscript, although VGF peptides are altered in a variety of disorders, the kinds of modified VGF peptides may differ according to the disease condition. VGF peptides could vary among them for their aa length and also specific bioactivity. Therefore, to highlight certain derived VGF peptides as early ALS diagnostic biomarkers, it is essential to precisely identify the VGF sequences changed in ALS.
Line 179: TLQP-9 is not introduced in Chapter 1.3. Why? Why is it presented separately?
Answer: The reason for this is that TLQP-9 is just the immunogen that was used to produce the TLQP antibody. The other TLQP peptides that were previously stated have instead been identified, and some of them were also tested to determine their bioactivity.
Lines 194-196: Is there any phenotypic/mechanistic/predictive significance to all these peptides?
Answer: Unfortunately, we are unable to speculate about the cause of the VGF peptide changes in ALS because most of the peptides altered are either never investigated for their biological activities, are only partially explored (sometimes with contradictory results), or are investigated in different pathways.
Line 228: “non-ALS controls” is not sufficient. Mention which.
Answer: We have adjusted the manuscript appropriately, adding the type of control subjects.
Lines 259 and 283: Which SOD1 mutant?
Answer: the SOD1-G93A transgenic mice, we have added this information into the manuscript
Lines 283-284: Which “VGF-derived peptide”?
Answer: TLQP-21, we have adjusted the manuscript appropriately
Line 286: The authors should acknowledge this is not really an ALS model.
Answer: We appreciate the referee's feedback and have adjusted the manuscript appropriately.
Line 293-295: Can you elaborate? Anything more concrete on the mechanism?
Answer: Unfortunately, the experiments were only done using vitality tests; hence, we are not able to hypothesize any mechanism
Line 297: Not really Conclusions. Rather a Discussion.
Answer: We have created a section for “discussion” and another for “conclusion”
Lines 280 and 282: SUNN8075 or SUN N8075?
Answer: it is SUN N8075
Comments on the Quality of English Language
Answer: We appreciate the referee's feedback for the quality of language and have adjusted the manuscript appropriately for all the words below
Line 49: “on a patient’s symptoms” done
Line 90: “routinary” – “routine” done
Line 129: should be plural “peptides” done
Line 134: “synapses” done
Line 204: “sensitivity” instead of “sensibility” done
Line 298: “emphasizes” should stand better than “raises”. done
Round 2
Reviewer 1 Report
Comments and Suggestions for Authors
First of all, thank you for the opportunity to review this manuscript again.
The relevance of this study is due to the insufficient number of studies devoted to the effectiveness of determining VGF and its derivative peptides in the diagnosis and prognosis of patients with ALS.
As I have already reported earlier, the title of the manuscript fully corresponds to the content.
In general, the introduction is written in a very interesting and highly scientific manner, using the latest scientific achievements in this field. However, changes to the introduction have added more clarity to this section.
In the revised version, the purpose clearly defined in a specific and brief form.
The revised version includes a separate section for materials indicating the methods used to compile this review, keywords and search engines.
The results are presented in clear language using visual illustrations to facilitate understanding of the text.
The changes in the structure of the discussion and conclusion were, in my opinion, successful. The main results were demonstrated in the discussion with explanations and comparisons with the results obtained in other works.
The conclusion turned out to be shorter and more specific